# No clear evidence for a domain-general violation of expectation effect in the pupillary responses of 9- to 10-month-olds

Christine Michel[1,2☯*], Miriam Langeloh[1,3☯], Markus R. Tünte[4], Moritz Köster[5], Stefanie Hoehl[1,4]

1 Max Planck Institute for Human Cognitive and Brain Sciences, Leipzig, Germany, 2 SRH University of Applied Sciences Heidelberg, Gera, Germany, 3 Institute of Psychology, Heidelberg University, Heidelberg, Germany, 4 Faculty of Psychology, University of Vienna, Austria, 5 University of Regensburg, Regensburg, Germany

☯ These authors contributed equally to this work.
* christine.michel@srh.de

## Abstract

Violation of expectation (VOE) paradigms are key to understanding infants' early knowledge. In VOE paradigms, infants are presented sequences of events either according with or violating regularities of their physical or social environment. Infants' violated expectations may result in a surprise response, such as longer looking times or specific neural correlates. There is an increasing interest in utilizing infants' pupil dilation as an index of their surprise. However, to date, no study has systematically examined infants' pupillary response across different VOE paradigms. In this preregistered study, we measured 9- to 10-month-olds' pupil dilation ($N=21$) in response to a common VOE paradigm across four knowledge domains (action, cohesion, number, solidity). In a pre-registered analysis, infants' pupillary response did not differ between expected and unexpected outcomes in any of these domains. We compared the effect of different analyses parameter choices in a specification curve analysis which revealed that very few choices would have led to significant results. The results demonstrate that across analytical decisions regarding data preprocessing and analysis we do not find evidence for the hypothesized effect. A subsequent permutation test revealed that our original data slightly diverges from randomly shuffled data. We can therefore not unambiguously reject the null hypothesis. We discuss these findings theoretically and methodologically and highlight the need for combining multiple measures to better understand the methods we apply to examine infants' knowledge about their environment.

## Introduction

From early on, human infants develop implicit knowledge about their physical and social environment [1,2]. This knowledge includes basic concepts of objects [3],

**Data availability statement:** Preprocessed data for the preregistered analysis of pupil dilation and looking times as well as the scripts and results of the specification curve analysisSCA are openly available on the Open Science Framework: https://osf.io/kwg9r/?view_only=676f9c9dabdf4d49a9f4d3e726e2bd3a.

**Funding:** This research was supported by the Max Planck Society (www.mpg.de). There was no additional external funding received for this study. The funders had no role in study design, data collection and analysis, decision to publish, or preparation of the manuscript.

**Competing interests:** The authors have declared that no competing interests exist.

others' actions [4] and numbers [5,6]. Infants' basic concepts guide and shape their early knowledge acquisition about their complex environment. For instance, observing events which violate infants' expectations based on their basic knowledge concepts increases infants' motivation to learn as indexed by increased exploration and hypothesis testing behavior [7].

The key to understanding infants' basic knowledge lies in their responses to events that are unlikely to happen in infants' daily experiences and therefore may violate basic expectations about outcomes of physical or social events (violation of expectation; VOE) [8]. Infants' differential responses to presumably expected vs. unexpected outcomes are interpreted as reflecting a surprise to unexpected events and, in consequence, as an understanding of basic concepts. Infants' VOE responses have long been assessed in looking time studies. For example, infants were presented with the unexpected event of a ball falling through a table, resulting in longer looking times compared to an expected outcome from 2 months onwards [3,9]. Similar results have been obtained in a multitude of subsequent studies (for an overview, see [10]).

Some authors have questioned the validity of looking time measures in VOE tasks, given the difficulty to control for potential confounds that may drive looking time differences between conditions (e.g., [11]). Yet, researchers have typically taken great effort to control for potential confounds, and as Stahl and Kibbe [12] pointed out, studies using other methods or even multi-methods approaches applied the VOE paradigm and often found corroborating evidence. One way of validating the looking times measure in VOE paradigms is the application of neurophysiological measures. In the past decades, neurophysiological research has linked infants' VOE responses to changes in brain activity such as event-related potentials (e.g., [13–15]), oscillatory dynamics [16–18] and entrained brain dynamics [19] in the electroencephalogram (EEG). Results of these studies show that the infant brain discriminates between familiar vs. unusual actions like turning a lamp on with the forehead or bringing a fork to the forehead [13–15,17,18], likely and unlikely outcomes in the number domain [16,18] as well as in the physical domain [18]. In most cases higher levels of brain activities have been reported for presumably unexpected vs. expected events. Thus, neurophysiological evidence supports the view that infants possess generative models of the world that are updated in response to prediction errors, i.e., VOE [2].

Recently, researchers have begun to apply pupil dilation as an additional approach for examining VOE responses in infants. Integrating predictive processing theories into the study of VOE responses provides a mechanistic framework for understanding how infants process violated expectations and how this processing may be reflected in dilated pupils. According to predictive processing, the brain constantly generates predictions about sensory input and updates generative models when discrepancies between expectation and perception occur [2]. Therefore, VOE could trigger physiological responses such as pupil dilation, reflecting cognitive effort required in the brain's attempt to update its internal model and, thus, learning [2,20].

In addition, pupil diameter has been described to increase to surprising or cognitively demanding events, indexing increased arousal, in adults and in infants [21–23]. Violated expectations may therefore introduce a state of surprise, which leads to

increased physiological arousals, detectable via dilated pupils. By integrating these domain-general theories, we can better understand the mechanistic link between expectation violations across various knowledge domains and pupillary responses. This could help to clarify how violated expectations in infants lead to observable physiological changes and why pupillometry may offer a viable, non-invasive measure of these processes.

Applying pupil dilation as an index of VOE may overcome a critical limitation associated with looking times, namely the possibility of measuring multiple trials while avoiding diminished responses over trials (for reviews see [24,25]). That is, habituation or fatigue in response to lengthy stimulus presentations in looking time studies limits the number of stimuli that can be presented, diminishing data quality (signal-to-noise ratio) and generalizability across stimulus types or knowledge domains. In addition, the pupillary diameter measures the physiological responses to VOE more directly than looking times. This allows to track dynamic changes in infants' processing, when compared to accumulated looking times. Compared to neurophysiological methods, which require at least the application of sensors (electrodes or optodes) on the scalp, the application of pupil dilation measurement is more feasible, easier and less intrusive. It is therefore of great interest to investigate whether pupillometry can be a valid measure for VOE processes in early infancy. The measurement of pupil size could offer an exciting experimental approach to examine infants' basic expectations in VOE paradigms.

First studies using pupil dilation to measure VOE responses show promising but mixed results, presumably reflecting the different topics of application and the relatively large age range of the participants in the different studies. On the one hand, infants as young as 4 months of age dilated their pupils to an irrational action in a social feeding context [26]. As feeding is something infants are very familiar with from early on, it may well be that infants build expectations about how feeding actions typically look like. Similarly, 8-month-olds responded with increased pupil dilation to unexpected physical events, such as a train changing color after emerging from a tunnel [27] and 11-month-olds' pupil dilated in response to unexpected social events [28]. Even in the auditive domain, deviant sounds led to pupil dilation in 14-month-olds [29]. On the other hand, Sirois and Jackson [30] could not find enhanced pupil dilation in response to an object disappearing behind a screen in an object permanence task in 10-month-olds (however, they do show that pupillometry can be a very valuable measure for disentangling ongoing information processing in a VOE-task) and the pupil of 9-month-olds did not change significantly in response to impossible body movements [31]. It is important to note that null results must be interpreted very cautiously. Based on null findings, we cannot conclude whether pupil dilation is a valid measure for VOE processes, but they still add important pieces of information to the literature. In addition, results of VOE-studies are of course dependent on the underlying concepts and their development. Pätzold and Liszkowski could show in two studies [32,33] that there are discriminative pupil dilation responses for unexpected events only in older infants (12 or 18 months) but not in younger infants aged 8 or 10 months who may not yet have developed the tested concepts. Here, pupil dilation seems to be a suitable measure to depict developmental trajectories of developing concepts.

Thus, results are mixed and the number of studies using pupil dilation in a VOE context is still limited and more systematic research is needed to thoroughly understand the pupillary response as a potential measure of infants' surprise in VOE paradigms. In particular, pupil dilation has not yet been applied in standard paradigms testing VOE in multiple core knowledge domains. In addition, there has been a critical debate in the field how much (arbitrary) decisions regarding the preprocessing and analysis of pupil dilation data impact the findings we obtain. Multiverse analyses allow for testing these effects [34,35].

One way to examine whether pupil dilatation is a valid measure in VOE paradigms is to apply a multi-methods approach and replicate existing findings [12]. Following this rationale, in the present study we examined pupil dilation as an index of infants' violated expectations across a variety of different knowledge domains. We aimed to investigate whether with the application of pupillometry, similar insights can be gained compared to neurophysiological measures but with an easier and more infant-friendly method. To test this idea, we analyzed pupil dilation data in response to a series of presumably expected and unexpected events across four knowledge domains (i.e., action, cohesion, number, and solidity) from a partially overlapping sample of 9- to 10-months-olds infants reported in Köster et al. [18]. This study found

a generalized VOE response to unexpected (vs. expected) outcomes reflected in a pronounced 4–5 Hz theta response in the infant EEG across all four knowledge domains.

Each trial consisted of three pictures, in which the first two pictures set the context for the final expected or unexpected outcome picture (Fig 1). Based on the assumption that pupil dilation captures infants' VOE to unexpected events [21,24], we hypothesized an increased pupil dilation in response to unexpected compared to expected outcome pictures. In addition, we analyzed infants' looking times to the outcome pictures (see S2 File). However, the short 5-second presentation time in a within-participants design is not standard in the field, so these results cannot be directly compared to previous studies. Based on prior research (e.g., [3,5]), we expected longer looking times for unexpected outcomes across all knowledge domains (see the preregistration at http://aspredicted.org/blind.php?x=5rx9b4).

## Materials and methods

### Participants

We tested 65 9- to 10-month-old infants. Three infants were excluded from further analyses due to technical problems with the eye tracker ($N = 1$) or unsuccessful calibration ($N = 2$). Furthermore, infants had to contribute at least two valid trials per domain to be included in the final sample. A total of 41 infants did not reach this criterion and were excluded from the analyses. The final sample consisted of 21 infants (13 female) with a mean age of 9 months and 31 days (age range 9.11–10.21 months and days). On average, infants contributed the following numbers of trials to the analysis per condition: 5.33 trials to the action expected condition, 5.43 trials to the action unexpected condition, 4.48 trials to the cohesion expected condition, 4.57 trials to the cohesion unexpected condition, 4.19 trials to the number expected condition, 4.29 to the number unexpected condition, 4.71 trials to the solidity expected condition and 5.24 trials to the solidity unexpected condition.

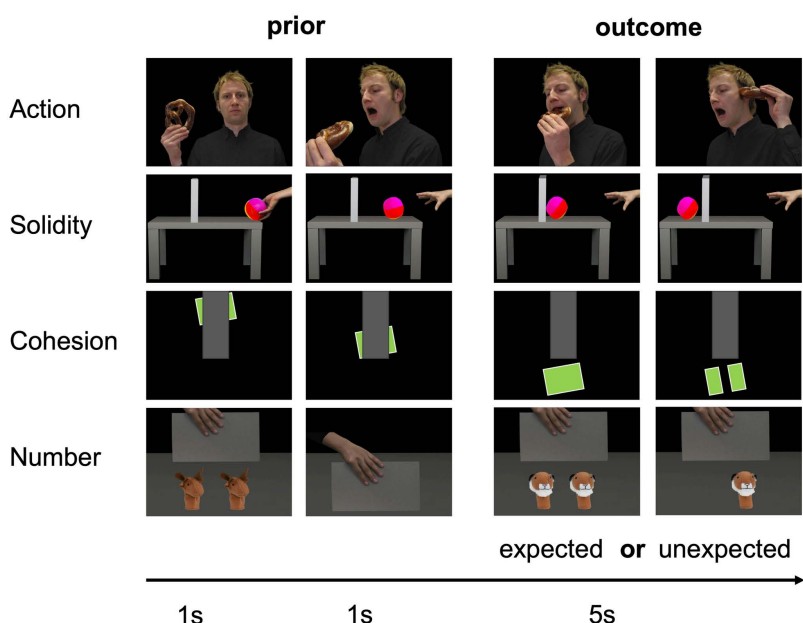

**Fig 1. Examples of trials for each domain and outcome.** Note. Sequence of a trial depicting the two priors and an expected or unexpected outcome. Figure adapted from Köster et al. (2019) (https://pmc.ncbi.nlm.nih.gov/articles/pmid/31603724/) published under the CC BY 4.0 License (https://creativecommons.org/licenses/by/4.0/). We adapted the duration of the stimulus presentation.

In addition to eye tracking, we simultaneously measured infants' EEG in 51 of the 65 tested infants and in 15 of the 21 infants in the final sample [18]. Preparing the EEG cap may have caused some fussiness in infants and took extra time before starting the experiment. This circumstance and the high number of 8 different conditions may explain the attrition rate of 63% (pupil dilation analysis), which is rather high for an eye tracking study but in line with another study measuring EEG and eye movements simultaneously [36]. We preregistered a final sample size of 30 infants. We had not checked data before we stopped data collection – as preregistered – at the end of February 2019. This is why we include only 21 (rather than 30) infants in the pupil dilation sample. All infants were born full-term (37–41 weeks of gestation) and without any known neurological problems. Informed verbal and written consent were obtained for each infant from one parent. The experiment was carried out in line with institutional protocols and the Declaration of Helsiniki and approved by the local ethics committee. Data collection took place at the Max Planck Institute for Human Cognitive and Brain Sciences in Leipzig (Germany) between 8[th] October 2018 and 28[th] February 2019.

## Stimuli, recording and presentation system

Stimuli consisted of static images. Stimuli were presented full screen on a 17" CRT monitor covering a visual angle of ca. 15.0° × 15.0°. Each trial consisted of a sequence of three pictures building upon each other. While the first two images (priors) set the context and presumably raised an expectation about what will be happening next, the third picture (outcome picture) depicted likely or unlikely outcomes of the sequence given the priors before. It is important to note that, in the following, we will refer to these events as "expected" and "unexpected", that is describing the psychological interpretation of the events which we aim to capture in our study. We analyzed infants' response to these outcome pictures. The brightness of the outcome pictures was balanced with Adobe Photoshop and thus the expected and unexpected outcome pictures did not differ, $t(3) = -1.24$, $p > 0.25$. Data were recorded on a Tobii X120 eye tracker with a sampling frequency of 60 Hz (Tobii Technology AB, Danderyd, Sweden) using Talk2Tobii version 1.1.0 (created by Luca Filipin, Fani Deligianni and Andrew T. Duchowski, http://psy.ck.sissa.it/t2t/About_T2T.html). Stimuli were presented using the Psychophysics Toolbox version 3.0.14 on Matlab version 9.3 [37,38].

## Procedure

The experiment took place inside a dimly lit and sound attenuated EEG booth. Infants sat on their parent's lap approximately 60 cm away from the monitor. In the final sample, 8 parents were asked to close their eyes or look down during the stimulus presentation and 13 parents watched the screen during stimulus presentation. As a result, we cannot entirely rule out the possibility that parental reactions to the stimuli may have influenced infant's responses. However, we consider it unlikely that such influences significantly and systematically affected infants' pupil dilation.

A 5-point-calibration accompanied with a lively music was performed using rotating yellow and blue stars as calibration stimuli. To assess the validity of the calibration, we presented two distinct pink spirals to the infants. The experimenter saw an overlay of the position of the spirals and infants' tracked gaze points and assessed whether infants' gaze points were accurately positioned on the spirals and not systematically shifted [39]. We re-calibrated infants if needed. The stimulus presentation started after a successful calibration was confirmed by the experimenter.

Fig 1 presents exemplary trial sequences for each domain. Each trial started with the presentation of a yellow duck (1000 ms) accompanied by a sound to attract infants' attention to the screen and a subsequent black screen with a random presentation duration between 500 ms and 700 ms. The priors were presented for 1000 ms each. The (un)expected outcome picture was presented for 5000 ms. We chose this timing to acquire enough trials for the EEG measurement and to allow the pupil to adjust to the presented stimuli [40]. In total, we presented 16 different sequences (4 different sequences per domain). Each sequence was presented two times in each of the conditions (expected vs. unexpected outcome), resulting in a total of 64 trials. Trials were presented in blocks of four trials of the same domain including two

expected and two unexpected outcomes. The blocks were separated by experimenter-controlled breaks. The order of the outcome (expected vs. unexpected) and the domains (action, cohesion, number, and solidity) were counterbalanced across all participants. Once the infant became fuzzy, the presentation was stopped.

### Data processing

Raw data were processed using TimeStudio [41]. In case both eyes were tracked successfully, data of both eyes were averaged. In case only one eye was tracked validly, only data of this eye were used. Only trials in which the infant fixated both priors and the outcome picture at least once were considered for further analyses. Therefore, data were filtered using an IVT-filter with a maximum angle between fixations of 0.5° and a minimum fixation duration of 200 ms [42]. Adjacent fixations were merged with a maximum gap of 75 ms. Data were filtered with a moving average of 5 samples and interpolated with a minimum interpolation gap of 10 samples [43]. As preregistered, in addition to the above-mentioned criterion of one fixation on each of the three pictures within a trial, infants had to provide valid data of at least 50% of the time of the presentation of the outcome picture to be included in the analysis [43]. Deviant from our preregistered analysis, we additionally rejected spurious data based on the second derivate. This additional processing led us to more valid data in the sense that single, spurious data points, which represented extreme values which most likely occurred due to errors in the measurement and not depicting the actual pupil size of the infant, were removed [40]. Data were baseline corrected using the first 500 ms of the outcome picture [43]. We averaged data within the time interval of 2 s – 3.5 s after the onset of the outcome picture. This time interval was chosen based on visual data inspection and previous studies using a similar time interval [40]. Finally, as preregistered, 15 trials of the final sample in total were excluded because their values differed more than 3 SD from the mean [43]. In accordance with our preregistration, we performed a 2 x 4 repeated measures analysis of variance (rmANOVA) with the within-subject factors outcome (expected vs. unexpected) and domain (action vs. cohesion vs. number vs. solidity) to test whether infants' pupils dilated more to unexpected outcomes as compared to expected outcomes. To test the impact of specific choices we made when preregistering our analysis, we performed a specification curve analysis [44]. With the help of a specification curve analysis, one can test how robust a result is across many reasonable analytical choices. We decided to perform this additional analysis to make sure our results are not driven by or only valid when taking specific choices in the process of data analysis. In our specification curve analysis, we therefore compared a multiverse of analysis choices, regarding the exclusion of (outlier) samples and trials, the interpolation and filter settings, baseline correction, the analyzed time window (additional time window 3.5s – 5s based on visual inspection), requirement of 2 valid trials per condition and domain, and the statistical analysis [34,35] (see S1 File for an overview of all specifications). In the specification curve analysis, we analyzed the results of all possible analyses that result from the combination of these decisions.

### Results

The 2 x 4 rmANOVA with the within-subject factors outcome (expected vs. unexpected) and domain (action vs. cohesion vs. number vs. solidity) only revealed a significant main effect of domain, $F(3,60) = 19.251$, $p < 0.001$, partial $\eta^2 = 0.490$. The main effect of outcome, $F(1,20) = 0.006$, $p = .940$, partial $\eta^2 = <.001$, and the interaction between domain and outcome, $F(3,60) = 0.688$, $p = 0.563$, partial $\eta^2 = 0.033$, were not significant, see Fig 2.

To explore differences between domains, we performed post-hoc pairwise $t$-tests on the averaged data of the expected and unexpected outcome for each domain (see Table 1). P-values were Bonferroni-corrected. Infants' pupils dilated the least in response to outcome pictures of the solidity domain ($M = −0.05$, $SD = 0.14$) as compared to any other domain. Pupil dilation in response to the outcomes of the action ($M = 0.10$, $SD = 0.12$), the cohesion ($M = 0.13$, $SD = 0.15$) and the number domain ($M = 0.07$, $SD = 0.13$) did not differ significantly, all $ps > 0.05$ (see Fig 3).

In the specification curve analysis, we compared 3058 different ways of analyzing the dataset (see Fig 4). Mirroring the results of our preregistered analysis, we find that most analyses performed in the specification curve analysis led to a

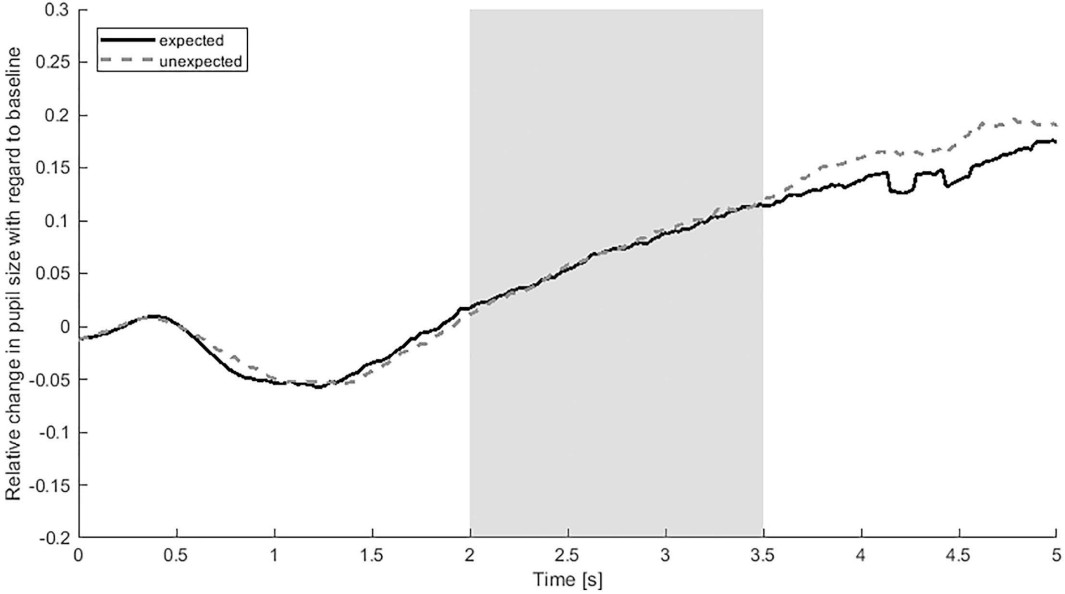

**Fig 2. Relative change in pupil size with regard to baseline over time in response to the expected and the unexpected outcomes.** Note. Average pupil dilation as compared to the baseline (0–500 ms after stimulus onset) over the course of the outcome picture (5 s). The solid black line represents responses to the expected outcomes averaged across domains, the grey dotted line to the unexpected outcomes averaged across domains. The grey rectangle marks the time interval used for the analysis.

**Table 1. Overview of the results of pairwise t-tests on the averaged data of the expected and unexpected outcome for each domain for the analysis of pupil dilation.**

|  | df | t | p | Cohen's d |
|---|---|---|---|---|
| Action vs. Cohesion | 20 | −1.44 | .98 | −0.31 |
| Action vs. Number | 20 | 1.98 | .37 | 0.43 |
| Action vs. Solidity | 20 | 5.80 | <.001 | 1.26 |
| Cohesion vs. Number | 20 | 2.57 | .11 | 0.56 |
| Cohesion vs. Solidity | 20 | −5.37 | <.001 | −1.17 |
| Number vs. Solidity | 20 | −4.48 | =.001 | −0.98 |

nonsignificant result ($N = 2853$, 93.30%). Only $N = 200$ analyses (6.54%) led to results confirming enhanced pupil dilation for unexpected outcomes and $N = 5$ (0.16%) to results showing enhanced pupil dilation for expected outcomes (see S1 File for more details and statistic of the specification curve analysis and the subsequent permutation test). The median effect of outcome on pupil dilation was .016, a very small effect.

Next, we compared the results of the specification curve analysis to results of a permutation test. We created 500 datasets for which the null hypothesis is true, by randomly shuffling outcome data (expected/unexpected) so that no match between outcome and pupil dilation can be present in the datasets. We repeated the specification curve analysis for each of the 500 randomly generated datasets. By comparing the results of the specification curve analysis from our original data and the shuffled data, we can estimate how (in)consistent our results are with showing no effect. In S1 File, we present a detailed description of the two indicators used: median estimate and share of significant results [44]. The median estimate ($Median_{specification\_curve} = 0.016$) of the specification curve analysis, including our preregistered data analysis path, is greater than in the randomly shuffled datasets ($Median_{permutation} = −0.000008$, $p < .001$). Further, the specification curve

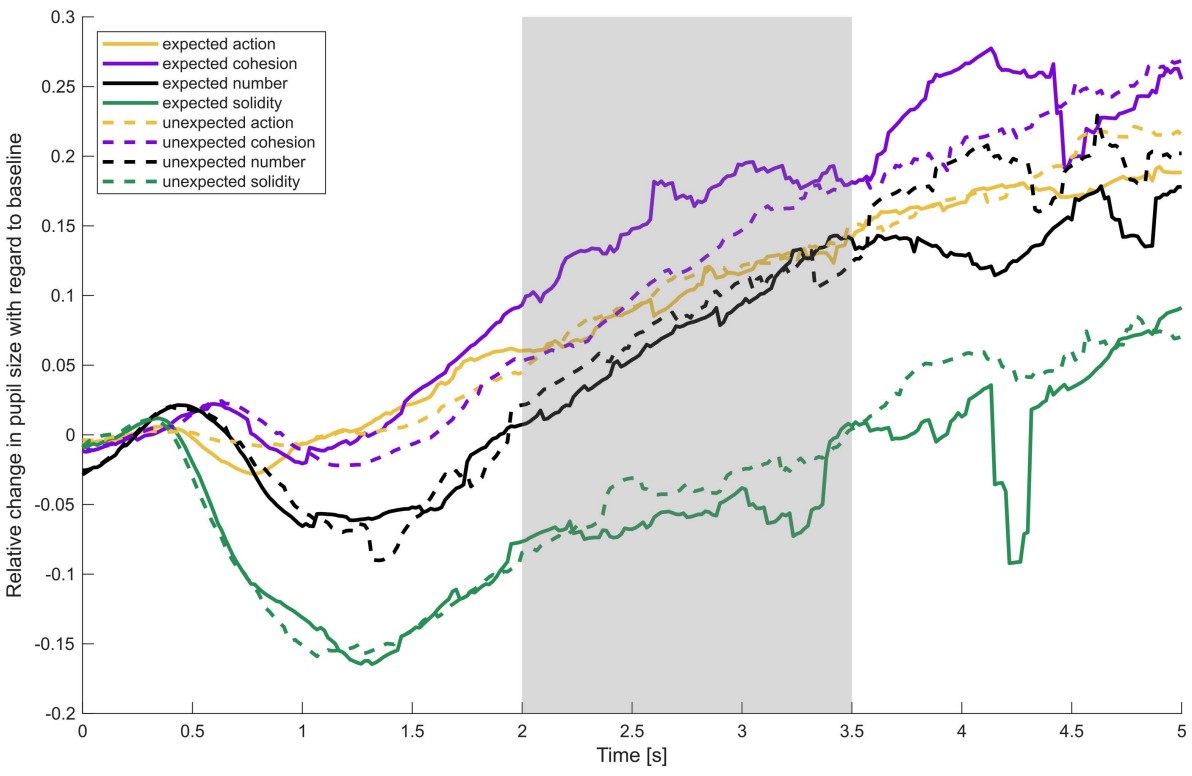

**Fig 3. Relative change in pupil size with regard to baseline over time in response to the expected and the unexpected outcomes for each domain.** Note. Mean change in pupil dilation, relative to the baseline (0–500 ms after stimulus onset) over the course of the outcome picture (5 s). Solid lines represent responses to the expected outcome, dotted lines to the unexpected outcome. The grey rectangle marks the time interval used for analysis.

analysis of our original data produced more significant ($N_{specification\_curve} = 205$) results than the shuffled data ($M_{shuffled} = 150.72$, $SD_{shuffled} = 26.63$, $p = 0.034$). Taking the results of this permutation test into account, we cannot unambiguously reject the null hypothesis as our results differ slightly from the permutation analysis.

## Discussion

In the present study, we aimed to elucidate whether pupil dilation captures VOE responses in 9- to 10-month-olds across four different knowledge domains (i.e., action, cohesion, number, and solidity). Our preregistered analysis did not provide evidence that pupil dilation varied between expected and unexpected outcomes in any of the four knowledge domains. The preregistered analysis choices to filter and interpolate data as well as to exclude trials with less than 50% valid data points or more than 3SD deviations from the mean were based on previous infant pupil dilation literature [42,43]. To further investigate which impact these analysis choices had, we performed a specification curve analysis. We did not find that specific decisions regarding the processing and analysis of data had an impact on the result. While the majority of analyses (93.30%) performed in the specification curve analysis confirmed the non-significant difference between expected and unexpected outcomes found in our preregistered analysis, it also showed that some analysis choices would have led to significant positive (6.54%) or negative (0.16%) relationships between outcome and pupil dilation.

Results of the permutation analysis showed that our data slightly diverges from shuffled data representing a null effect. However, the median effect size estimate of our data is still very small (median estimate = 0.016), meaning

**A**

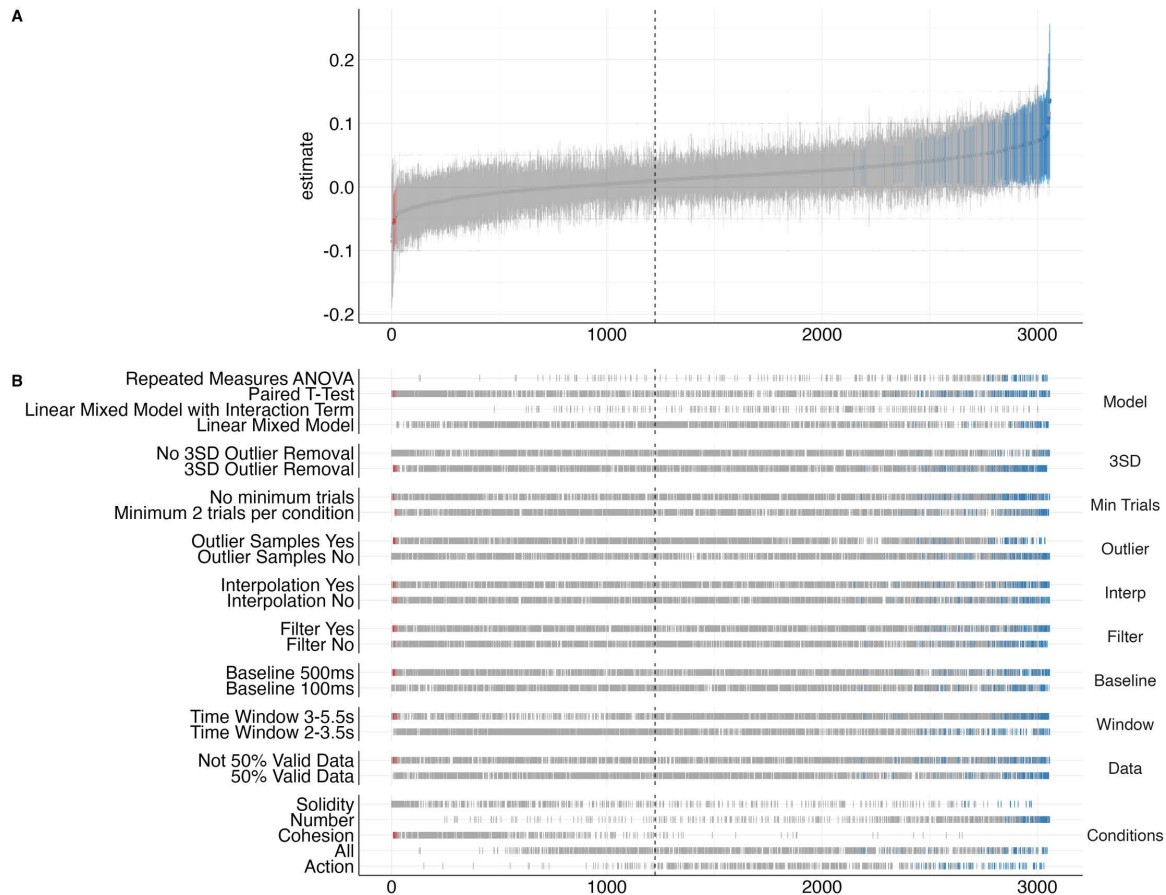

**Fig 4. Results of the specification curve analysis.** Note. In A) the specification curve is displayed with specifications (x-axis) ordered by standardized beta regression estimates and corresponding 95% confidence interval (y-axis). Red color indicates a significant negative result, blue color a significant positive result, and grey color a non-significant result. Our preregistered analysis is marked by the dashed black line. In B) the results from the specification curve analysis are displayed separate for each analytical decision.

that any effect of the expected or unexpected outcome on pupil dilation is potentially too unreliable to be detected in smaller sample sizes such as the present study. In addition, the meaningfulness of such a small effect remains questionable.

To investigate domain-general pupillary responses, we applied stimuli that previously elicited distinct neural responses to expected and unexpected outcomes in 9-month-old infants and we used the exact same timing of the presentation sequence as in the companion EEG study [18]. The presentation time of the outcome picture was chosen based on previous literature showing that the pupil can adjust to a stimulus within 5 s already in younger infants [26,39,40,45]. Thus, we designed our study in a way to capture possible VOE responses in the pupillary dilation of 9- to 10-month-olds. If anything, we only found a very small effect in the permutation analysis of pupil dilation between expected and unexpected outcomes across various knowledge domains. This adds a critical test of concept to previous pupil dilation VOE studies, which have reported mixed results: Some former studies did not demonstrate discriminative pupil dilation in response to a potential violation of expectation at this age [30–33]. In some of those studies, the authors found a pupillary surprise response to unexpected events in the second year of life [32,33]. In contrast, other studies demonstrated enhanced pupil size [27] in even younger infants [26].

There are several possible interpretations and explanations for our results. First, it could be that domain-general VOE effects – the ability to generalize surprise across various knowledge domains (e.g., action, cohesion, number, solidity) – may not be fully developed at this age. According to predictive processing theories, discrepancies between expectation and reality (i.e., VOE) could trigger physiological responses like dilated pupils. It may be the case that the VOE effects elicited by our stimuli were not strong enough to be reflected in the measure of pupil dilation, for instance, because infants have not yet built robust generative models in all of these domains or because they give relatively more weight to novel sensory input than their emerging priors. However, the same stimuli presented here have formerly elicited a VOE response in the EEG of infants at the same age [18,19], which makes us confident that the stimuli can elicit a VOE response.

Another potential explanation for the non-significant effect (or if anything only very weak effect) of outcome on pupil dilation could be that pupil size in younger infants during their first postnatal year of life does not respond to non-emotional, cognitive VOE events but rather to (socially) emotionally arousing events [24,39,40]. Pupil size is controlled via the noradrenergic system [23], and the involvement of the noradrenergic system may be low in acquiring basic physical and social concepts at this age. Potentially, this may prevent young infants from being overwhelmed or stressed by the realm of novel information they sample about their environment. Speculatively, emotionally arousing stimuli could elicit a noradrenergic response linked to neurophysiological arousal earlier in development, compared to more neutral physical and social stimuli. Given the mixed results in pupil dilation in infants between 4 and 18 months of age, further studies with increased sample sizes are needed. Therefore, one interesting aspect could be to compare the differential responses to non-emotional, cognitive surprise events and affectively arousing events.

Finally, pupillometry may not be a robust and reliable index for VOE responses in 9- to 10-month-olds. Despite showing that pupillary responses are often linked to expectation violations in older infants and in other domains, our study did not capture robust VOE effects in younger infants in domain-general stimuli. Critically, we did not find strong support for our hypothesis and non-significant results can only be interpreted tentatively.

We also analyzed infants' looking times to the screen during the presentation of the outcome picture, but we did not find differences between expected and unexpected outcomes (see S2 File). Possibly, the presentation duration of 5 s, which we optimized for measuring pupil dilation while at the same time allowing us to present many trials, was too short to capture VOE responses in infants' looking times. This idea is in line with studies using an even longer presentation duration, for example Stahl and Feigenson [7], without finding differences in looking times between expected and unexpected events (but see [46] who found an effect of VOE in looking times showing the outcome picture for only 3s).

In our paper, we used a specification curve analysis to shed light on the null findings in our preregistered analysis [34,35,44]. This approach proved to be a promising tool not only to clarify non-significant results, but also to capture the complexity of the data. Interestingly, we find that our preregistered analysis did not indicate a significant relationship between outcome and pupil dilation. However, the specification curve analysis revealed that our results slightly differ from results expected given no relationship between outcome and pupil dilation. Thus, the specification curve analysis may be a helpful tool for future (infant) studies, not only but also on pupil dilation, as they can play an important role in interpreting null results.

To summarize, we could not find any empirical evidence for VOE responses in 9- to 10-month-olds analyzing pupil dilation in our main analysis. The deviation of our data from the shuffled data in the permutation test of the specification curve analysis suggests, if anything, only a weak effect. Our findings recommend that while pupillometry offers methodological advantages, it may not serve as a reliable standalone indicator of domain-general VOE responses in infants under one year. This, and keeping in mind that the absence of evidence for a strong effect does not equate to evidence of absence, calls for caution in interpreting null or inconsistent pupil-based findings in early infancy and underscores the need for multi-modal approaches across age [12]. One promising approach remains the assessment of several VOE markers simultaneously and thereby the cross-validation of different measures (e.g., neural, physiological and behavioral assessments).

However, this is methodologically highly difficult in infancy research. Our findings highlight that, given their primacy role in our understanding of infants' basic concepts of their world, it is a central challenge to further validate and get a better grip on neurocognitive and behavioral markers of infants' predictions about their physical and social environment.

## Supporting information

**S1 File. Specification Curve Analysis on pupil dilation.**
(DOCX)

**S2 File. Analysis of looking time data.**
(DOCX)

## Acknowledgments

We are grateful to the infants and parents who participated.

## Author contributions

**Conceptualization:** Christine Michel, Miriam Langeloh, Moritz Köster, Stefanie Hoehl.

**Data curation:** Christine Michel, Miriam Langeloh, Markus R. Tünte, Moritz Köster.

**Formal analysis:** Christine Michel, Miriam Langeloh, Markus R. Tünte.

**Funding acquisition:** Stefanie Hoehl.

**Investigation:** Christine Michel, Miriam Langeloh.

**Methodology:** Christine Michel, Miriam Langeloh, Moritz Köster, Stefanie Hoehl.

**Project administration:** Stefanie Hoehl.

**Resources:** Stefanie Hoehl.

**Software:** Christine Michel, Moritz Köster.

**Supervision:** Stefanie Hoehl.

**Validation:** Christine Michel.

**Visualization:** Christine Michel, Markus R. Tünte.

**Writing – original draft:** Christine Michel, Miriam Langeloh.

**Writing – review & editing:** Christine Michel, Miriam Langeloh, Markus R. Tünte, Moritz Köster, Stefanie Hoehl.

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
