## [Decision Letter · Decision Letter 0]

10 Jul 2025

Dear Dr. Michel,

Thank you for submitting your manuscript to PLOS ONE. After careful consideration, we feel that it has merit but does not fully meet PLOS ONE’s publication criteria as it currently stands. Therefore, we invite you to submit a revised version of the manuscript that addresses the points raised during the review process.

We look forward to receiving your revised manuscript.

Kind regards,

Tomasz W. Kaminski

Academic Editor

PLOS ONE

Journal Requirements:

[This research was supported by the Max Planck Society (www.mpg.de)].

[This research was supported by the Max Planck Society. We are grateful to the infants and parents who participated.]

[This research was supported by the Max Planck Society (www.mpg.de)].

4. We note that Figure 1 includes an image of a participant in the study.

Additional Editor Comments:

Dear Authors,

Thank you for your patience during the review process. I apologize for the delay - the topic is highly specialized, and it took time to secure input from the appropriate experts in the field.

Both reviewers have now submitted their comments, and I’m pleased to inform you that both recommend minor revisions, which is also in line with my own assessment of the manuscript.

At this point, I invite you to revise your manuscript by addressing the reviewers’ comments carefully and thoroughly. Once your revised version is submitted, we will proceed with the next steps.

Please don’t hesitate to reach out if you have any questions.

Best regards,

Tomasz W. Kaminski

Reviewers' comments:

Reviewer's Responses to Questions

**Comments to the Author**

1. Is the manuscript technically sound, and do the data support the conclusions?

Reviewer #1: Partly

Reviewer #2: Yes

2. Has the statistical analysis been performed appropriately and rigorously?

Reviewer #1: Yes

Reviewer #2: Yes

3. Have the authors made all data underlying the findings in their manuscript fully available?

Reviewer #1: Yes

Reviewer #2: Yes

4. Is the manuscript presented in an intelligible fashion and written in standard English?

Reviewer #1: Yes

Reviewer #2: Yes

Reviewer #1: The authors investigate whether pupil dilation in infants serves as a marker of expectation violation (VOE) by presenting four different paradigms. They hypothesized that unexpected events would elicit greater pupil dilation. The results did not support this hypothesis and thus suggests that pupil dilation may not be a good way to estimate VOE. Overall, the manuscript addresses a relevant question and is well written. There are several methodological ambiguities and analytical choices that require clarification or further justification. In addition I suggest a few minor edits

1. Lack of Support for Hypothesis and Framing of Results:

The authors conclude that they did not find “strong evidence” for pupil dilation during VOE. However, the data rather suggest evidence against such an effect. The wording in the conclusion should be revised to reflect this interpretation, while acknowledging that the absence of evidence is not evidence of absence (which the authors already do)

2. Unclear Participant Breakdown and EEG Subsample:

The authors should clarify the samples used for the different analyses. How many infants participated in the VOE, how many wore EEG caps, was there an overlap? Right now the authors write that the sample included 65 infants and that 3 were excluded. But then they write that the pre-registered protocol included 30, but that only 21 were sampled. I am confused.

3. Justification of Analytical Methods:

I am not familiar with the curve analysis applied in addition to the pre-registered mixed ANOVA is not a standard approach in this context. The rationale for using this analysis should be explained more clearly, especially since it forms a central part of the authors’ interpretation.

4. Effect Sizes Missing:

The manuscript would benefit from consistent inclusion of effect size estimates to help contextualize the findings and support claims of null or small effects.

5. Table 1 Redundancy:

The information in Table 1 could be integrated into the main text, as it currently seems superfluous.

6. Figure Legibility:

Several figures could benefit from higher resolution and adjustment of fonts. For figure 4 the font seems too small and figure 3 seems very pixelated. (Perhaps this is only in this version?).

7. Stimulus Visibility to Parents:

It would be helpful to clarify whether the sunglasses used to occlude parental vision fully prevented them from seeing the stimuli, as this may affect the infant’s behavior.

Reviewer #2: Dear Authors,

I would like to congratulate you on a novel and well-executed study. Your preregistered and methodologically rigorous investigation addresses an important and timely question regarding the suitability of pupil dilation as a marker of violation-of-expectation (VOE) responses in infancy.

Below, I offer a few substantive suggestions that I believe could further strengthen your manuscript:

1. Strengthen the theoretical rationale for expecting domain-general VOE effects in pupil dilation

While you refer to previous EEG findings (Köster et al., 2018) and arousal-based explanations, the theoretical basis for expecting domain-general effects in pupil dilation remains underdeveloped. I suggest expanding on why pupil dilation should be sensitive to violations across diverse knowledge domains (e.g., action, cohesion, number, solidity), potentially by integrating predictive processing theories or broader arousal/surprise frameworks in infancy. A clearer mechanistic link between expectation violation and pupillary response would help frame your study's goals and findings.

2. Add a clearer and more detailed description of the statistical analysis

The manuscript currently lacks a concise summary of the statistical analysis pipeline in the main text. While the specification curve is discussed in the results and supplemental sections, the main paper would benefit from a clearer outline of the core analytical steps and decision points, particularly regarding how the pupil dilation data were processed, analyzed, and interpreted. This would enhance transparency and reproducibility.

3. Clarify the theoretical implications of the null result

The discussion could more fully engage with what the null findings mean for the field. I recommend addressing two possible interpretations:

(a) that pupil dilation may not serve as a reliable index of VOE at 9-10 months, and/or

(b) that domain-general VOE effects themselves may be less robust or not yet fully developed at this age.

Framing your null results in the context of broader theories of cognitive surprise, arousal, or predictive learning in infancy would increase the theoretical value of your findings and avoid the perception of inconclusiveness.

4. Provide a clearer take-home message

The manuscript would benefit from a more conclusive final paragraph that distills the contribution of your findings. For instance, you might emphasize that while pupillometry is a promising tool, it may not reliably detect domain-general VOE effects in infancy, at least under the current task parameters. Consider adding a statement such as:

“Our findings suggest that while pupillometry offers methodological advantages, it may not serve as a reliable standalone indicator of domain-general VOE responses in infants under one year. This calls for caution in interpreting null or inconsistent pupil-based findings in early infancy and underscores the need for multi-modal approaches.”

Again, I commend the authors for a carefully designed and transparently reported study. I hope these suggestions are constructive and help enhance the clarity and theoretical contribution of the manuscript.

Sincerely,

Reviewer

**Do you want your identity to be public for this peer review?** For information about this choice, including consent withdrawal, please see our Privacy Policy

Reviewer #1: **Yes: ** Anders Rasmussen

Reviewer #2: **Yes: ** Wolosowicz Marta

---

## [Author Response · Author response to Decision Letter 1]

2 Sep 2025

Dear Dr. Tomasz W. Kaminski

Dear Reviewers,

we thank the Reviewers for their thoughtful comments. We believe the manuscript has improved, especially regarding conclusions drawn from our data. Please find below the reviewers’ comments in italic and our responses in non-italic. Page numbers refer to the manuscript with tracked changes.

We thank you for considering our manuscript PONE-D-25-21638 for publication in PLOS ONE.

Reviewer 1’s comments

Comment 1: The authors investigate whether pupil dilation in infants serves as a marker of expectation violation (VOE) by presenting four different paradigms. They hypothesized that unexpected events would elicit greater pupil dilation. The results did not support this hypothesis and thus suggests that pupil dilation may not be a good way to estimate VOE. Overall, the manuscript addresses a relevant question and is well written. There are several methodological ambiguities and analytical choices that require clarification or further justification. In addition I suggest a few minor edits

1. Lack of Support for Hypothesis and Framing of Results:

The authors conclude that they did not find “strong evidence” for pupil dilation during VOE. However, the data rather suggest evidence against such an effect. The wording in the conclusion should be revised to reflect this interpretation, while acknowledging that the absence of evidence is not evidence of absence (which the authors already do)

We framed the results as no “strong” evidence as the permutation test of the specification curve analysis showed that our results slightly differed from the permutation analysis analyzing shuffled data (i.e., data representing data valid under the assumption of the null hypothesis). We therefore cannot unambiguously reject the null hypothesis and we are reluctant to state that our data only suggests evidence against an effect of VOE on pupil dilation. However, we agree with the reviewer that our results can be stated more clearly in the conclusion.

In accordance with comment 4 of Reviewer 2, we have now changed the conclusion section of the manuscript as follows:

“To summarize, we could not find any empirical evidence for VOE responses in 9- to 10-month-olds analyzing pupil dilation in our main analysis. The deviation of our data from the shuffled data in the permutation test of the specification curve analysis suggests, if anything, only a weak effect. Our findings recommend that while pupillometry offers methodological advantages, it may not serve as a reliable standalone indicator of domain-general VOE responses in infants under one year. This, and keeping in mind that the absence of evidence for a strong effect does not equate to evidence of absence, calls for caution in interpreting null or inconsistent pupil-based findings in early infancy and underscores the need for multi-modal approaches across age (1). One promising approach remains the assessment of several VOE markers simultaneously and thereby the cross-validation of different measures (e.g., neural, physiological and behavioral assessments). However, this is methodologically highly difficult in infancy research. Our findings highlight that, given their primacy role in our understanding of infants’ basic concepts of their world, it is a central challenge to further validate and get a better grip on neurocognitive and behavioral markers of infants' predictions about their physical and social environment.”

(page 20)

Comment 2: Unclear Participant Breakdown and EEG Subsample:

The authors should clarify the samples used for the different analyses. How many infants participated in the VOE, how many wore EEG caps, was there an overlap? Right now the authors write that the sample included 65 infants and that 3 were excluded. But then they write that the pre-registered protocol included 30, but that only 21 were sampled. I am confused.

We have now clarified sample characteristics in the manuscript and additionally integrated the information from Table 1 in the main text (see Comment 5 of Reviewer 1). The manuscript now states the following:

“We tested 65 9- to 10-month-old infants. Three infants were excluded from further analyses due to technical problems with the eye tracker (N = 1) or unsuccessful calibration (N = 2). Furthermore, infants had to contribute at least two valid trials per domain to be included in the final sample. A total of 41 infants did not reach this criterion and were excluded from the analyses. The final sample consisted of 21 infants (13 female) with a mean age of 9 months and 31 days (age range 9.11 – 10.21). On average, infants contributed the following numbers of trials to the analysis per condition: 5.33 trials to the action expected condition, 5.43 trials to the action unexpected condition, 4.48 trials to the cohesion expected condition, 4.57 trials to the cohesion unexpected condition, 4.19 trials to the number expected condition, 4.29 to the number unexpected condition, 4.71 trials to the solidity expected condition and 5.24 trials to the solidity unexpected condition. In addition to eye tracking, we simultaneously measured infants’ EEG in 51 of the 65 tested infants and in 15 of the 21 infants in the final sample (18). (page 8)

Comment 3: Justification of Analytical Methods:

I am not familiar with the curve analysis applied in addition to the pre-registered mixed ANOVA is not a standard approach in this context. The rationale for using this analysis should be explained more clearly, especially since it forms a central part of the authors’ interpretation.

We now clarify our decision to run a specification curve analysis in addition to the rm-ANOVA in more detail in the manuscript:

“To test the impact of specific choices we made when preregistering our analysis, we performed a specification curve analysis (3). With the help of a specification curve analysis, one can test how robust a result is across many reasonable analytical choices. We decided to perform this additional analysis to make sure our results are not driven by or only valid when taking specific choices in the process of data analysis. In our specification curve analysis, we therefore compared a multiverse of analysis choices, regarding the exclusion of (outlier) samples and trials, the interpolation and filter settings, baseline correction, the analyzed time window (additional time window 3.5s – 5s based on visual inspection), requirement of 2 valid trials per condition and domain, and the statistical analysis (4,5) (see Supplementary Material S1 for an overview of all specifications ). In the specification curve analysis, we analyze the results of all possible analyses that result from the combination of these decisions.”

(page 12)

Comment 4: Effect Sizes Missing:

The manuscript would benefit from consistent inclusion of effect size estimates to help contextualize the findings and support claims of null or small effects.

The effect sizes partial η² (page 12) and Cohen’s d (Table 2, page 14) are included in the manuscript.

Comment 5: Table 1 Redundancy:

The information in Table 1 could be integrated into the main text, as it currently seems superfluous.

We have now integrated the information from Table 1 in the main text, see also Comment 2 of Reviewer 1.

Comment 6: Figure Legibility:

Several figures could benefit from higher resolution and adjustment of fonts. For figure 4 the font seems too small and figure 3 seems very pixelated. (Perhaps this is only in this version?).

We have ameliorated the quality of Figure 3.

We have enlarged the font of Figure 4.

Comment 7: Stimulus Visibility to Parents:

It would be helpful to clarify whether the sunglasses used to occlude parental vision fully prevented them from seeing the stimuli, as this may affect the infant’s behavior.

We thank the reviewer for asking for clarity with regard to the sunglasses. Because of the reviewer’s comment, we went back to the videos of the participants. We observed that, unlike the standard procedure in our laboratory, parents in this study did not wear sunglasses. Instead, many—though not all—were instructed to either look down or close their eyes during the stimulus presentation. As a result, we cannot entirely rule out the possibility that parental reactions to the stimuli may have influenced infant’s responses. However, we consider it unlikely that such influences significantly and systematically affected infant pupil dilation. We now clarify this issue in the main manuscript:

“In the final sample, 8 parents were asked to close their eyes or look down during the stimulus presentation and 13 parents watched the screen during stimulus presentation. As a result, we cannot entirely rule out the possibility that parental reactions to the stimuli may have influenced infant’s responses. However, we consider it unlikely that such influences significantly and systematically affected the infant pupil dilation.”

(page 10)

Reviewer 2

Comment 1: I would like to congratulate you on a novel and well-executed study. Your preregistered and methodologically rigorous investigation addresses an important and timely question regarding the suitability of pupil dilation as a marker of violation-of-expectation (VOE) responses in infancy. Below, I offer a few substantive suggestions that I believe could further strengthen your manuscript:

Strengthen the theoretical rationale for expecting domain-general VOE effects in pupil dilation. While you refer to previous EEG findings (Köster et al., 2018) and arousal-based explanations, the theoretical basis for expecting domain-general effects in pupil dilation remains underdeveloped. I suggest expanding on why pupil dilation should be sensitive to violations across diverse knowledge domains (e.g., action, cohesion, number, solidity), potentially by integrating predictive processing theories or broader arousal/surprise frameworks in infancy. A clearer mechanistic link between expectation violation and pupillary response would help frame your study's goals and findings.

We thank the reviewer for this helpful perspective. We have now incorporated these ideas in the Introduction:

“Recently, researchers have begun to apply pupil dilation as an additional approach for examining VOE responses in infants. Integrating predictive processing theories into the study of VOE responses provides a mechanistic framework for understanding how infants process violated expectations and how this processing may be reflected in dilated pupils. According to predictive processing, the brain constantly generates predictions about sensory input and updates generative models when discrepancies between expectation and perception occur (6). Therefore, VOE could trigger physiological responses such as pupil dilation, reflecting cognitive effort required in the brain's attempt to update its internal model and, thus, learning (6,7).

In addition, pupil diameter has been described to increase to surprising or cognitively demanding events, indexing increased arousal, in adults and in infants (8–10). Violated expectations may therefore introduce a state of surprise, which leads to increased physiological arousals, detectable via dilated pupils. By integrating these domain-general theories, we can better understand the mechanistic link between expectation violations across various knowledge domains and pupillary responses. This could help to clarify how violated expectations in infants lead to observable physiological changes and why pupillometry may offer a viable, non-invasive measure of these processes.

Applying pupil dilation as an index of VOE may to overcome a critical limitation associated with looking times, namely the possibility of measuring multiple trials while avoiding diminished responses over trials (for reviews see 11,12).“

(page 4 & 5)

Comment 2: Add a clearer and more detailed description of the statistical analysis

The manuscript currently lacks a concise summary of the statistical analysis pipeline in the main text. While the specification curve is discussed in the results and supplemental sections, the main paper would benefit from a clearer outline of the core analytical steps and decision points, particularly regarding how the pupil dilation data were processed, analyzed, and interpreted. This would enhance transparency and reproducibility.

We now describe the analysis of looking times in more detail in the supplements S2. Core analytical steps of the pupil dilation data and the statistical analysis pipeline are outlined in detail in the section “Data processing” in the main manuscript, starting on page 11.

Comment 3: Clarify the theoretical implications of the null result

The discussion could more fully engage with what the null findings mean for the field. I recommend addressing two possible interpretations:

(a) that pupil dilation may not serve as a reliable index of VOE at 9-10 months, and/or

(b) that domain-general VOE effects themselves may be less robust or not yet fully developed at this age.

Framing your null results in the context of broader theories of cognitive surprise, arousal, or predictive learning in infancy would increase the theoretical value of your findings and avoid the perception of inconclusiveness.

We agree with the reviewer that there are several possible explanations for the lack of clear results in our study. We rewrote the paragraph in the Discussion section as follows:

“There are several possible interpretations and explanations for our results. First, it could be that domain-general VOE effects – the ability to generalize surprise across various knowledge domains (e.g., action, cohesion, number, solidity) – may not be fully developed at this age. According to predictive processing theories, discrepancies between expectation and reality (i.e., VOE) could trigger physiological responses like dilated pupils. It may be the case that the VOE effects elicited by our stimuli were not strong enough to be reflected in the measure of pupil dilation, for instance, because infants have not yet built robust generative models in all these domains or because they give relatively more weight to novel sensory input than their emerging priors. However, the same stimuli presented here have formerly elicited a VOE response in the EEG of infants at the same age (2,13), which makes us confident that the stimuli can elicit a VOE response.

Another potential explanation for the non-significant effect (or if anything only very weak effect) of outcome on pupil dilation could be that pupil size in younger infants during their first postnatal year of life does not respond to non-emotional, cognitive VOE events but rather to (socially) emotionally arousing events (11,14,15). Pupil size is controlled via the noradrenergic system (10), and the involvement of the noradrenergic system may be low in acquiring basic physical and social concepts at this age. Potentially, this may prevent young infants from being overwhelmed or stressed by the realm of novel information they sample about their environment. Speculatively, emotionally arousing stimuli could elicit a noradrenergic response linked to neurophysiological arousal earlier in development, compared to more neutral physical and social stimuli. Given the mixed results in pupil dilation in infants between 4 and 18 months of age, further studies with increased sample sizes are needed. Therefore, one interesting aspect could be to compare the differential responses to non-emotional, cognitive surprise events and affectively arousing events.

Finally, pupillometry may not be a robust and reliable index for VOE responses in 9- to 10-month-olds. Despite showing that pupillary responses are often linked to expectation violations in older infants and in other domains, our study did not capture robust VOE effects in younger infants in domain-general stimuli. Critically, we did not find strong support for our hypothesis and non-significant results can only be interpreted tentatively.”

(page 18 & 19)

Comment 4: Provide a clearer take-home message

The manuscript would benefit from a more conclusive final paragraph that distills the contribution of your findings. For instance, you might emphasize that while pupillometry is a promising tool, it may not reliably detect domain-general VOE effects

---

## [Editor Report · Decision Letter 1]

3 Sep 2025

No clear evidence for a domain-general violation of expectation effect in the pupillary responses of 9- to 10-month-olds

PONE-D-25-21638R1

Dear Dr. Michel,

We’re pleased to inform you that your manuscript has been judged scientifically suitable for publication and will be formally accepted for publication once it meets all outstanding technical requirements.

Kind regards,

Tomasz W. Kaminski

Academic Editor

PLOS ONE